# Randomized, observer-blind, controlled Phase 1 study of the safety and immunogenicity of the *Na*-GST-1/Alhydrogel hookworm vaccine with or without a CpG ODN adjuvant in hookworm-naïve adults

David J. Diemert [1,2*], Maria Zumer[1], Mark Bova[3], Christina Gibbs-Tewary[1], Elissa M. Malkin[1], Doreen Campbell[1], Lara Hoeweler[2], Guangzhao Li[2], Maria Elena Bottazzi[4,5], Peter J. Hotez[4,5], Jeffrey M. Bethony[2]

**1** Department of Medicine, School of Medicine and Health Sciences, The George Washington University, Washington, DC, United States of America, **2** Department of Microbiology, Immunology and Tropical Medicine, School of Medicine and Health Sciences, The George Washington University, Washington, DC, United States of America, **3** Department of Epidemiology, Milken Institute School of Public Health, The George Washington University, Washington, DC, United States of America, **4** Division of Pediatric Tropical Medicine, Department of Pediatrics, National School of Tropical Medicine, Texas Children's Hospital Center for Vaccine Development, Baylor College of Medicine, Houston, Texas, United States of America, **5** Department of Molecular Virology and Microbiology, Baylor College of Medicine, Houston, Texas, United States of America

* ddiemert@gwu.edu

## Abstract

### Background

Recombinant *Necator americanus* Glutathione-S-Transferase-1 (*Na*-GST-1) formulated on Alhydrogel (*Na*-GST-1/Alhydrogel) is being developed to prevent anemia and other complications of *N. americanus* infection. Antibodies induced by vaccination with recombinant *Na*-GST-1 are hypothesized to interfere with the blood digestion pathway of adult hookworms in the host. Phase 1 trials have demonstrated the safety of *Na*-GST-1 formulated on Alhydrogel, but further optimization of the vaccine adjuvant formulation may improve humoral immune responses, thereby increasing the likelihood of vaccine efficacy.

### Methods

A randomized, observer-blind, dose escalation Phase 1 trial was conducted in 24 healthy, hookworm-naïve adults. In each cohort of 12 participants, 4 were randomized to receive 100 μg of *Na*-GST-1/Alhydrogel and 8 to receive 30 μg or 100 μg of *Na*-GST-1/Alhydrogel plus the Cytosine-phospho-Guanine (CpG) oligodeoxynucleotide Toll-like receptor-9 agonist, CpG 10104, in the first and second cohorts, respectively. Progression to the second cohort was dependent upon evaluation of 7-day safety data after all participants in the first cohort had received the first dose of vaccine. Three intramuscular injections of study product were administered on days 0, 56, and 112, after which participants were followed

**Data availability statement:** De-identified individual participant data that underlie the results reported in this Article are available to anyone who wishes to access the data from the time of publication with no end date, for any purpose. The complete study data are available online at https://gwvru.smhs.gwu.edu/clinical.

**Funding:** This project has been funded with an award from the Ministry of Foreign Affairs of the Government of the Netherlands (grant #23386). The Principal Investigator of this grant was PJH. The funder had no role in study design, data collection and analysis, decision to publish, or preparation of the manuscript.

**Competing interests:** I have read the journal's policy and the authors of this manuscript have the following competing interests: PJH, MEB, JMB, and DJD are named as inventors on a patent for a multivalent helminth vaccine (US8211438B2). All other authors declare no competing interests.

for 6 months. IgG and IgG subclass antibody responses to *Na*-GST-1 were measured by qualified indirect ELISAs at pre- and post-vaccination time points.

## Results

*Na*-GST-1/Alhydrogel administered with or without CpG 10104 was well-tolerated. The most common solicited adverse events were mild injection site tenderness and pain, and mild headache. There were no vaccine-related serious adverse events or adverse events of special interest. Both dose concentrations of *Na*-GST-1/Alhydrogel plus CpG 10104 had significantly higher post-vaccination levels of antigen-specific IgG antibody compared to *Na*-GST-1/Alhydrogel without CpG, starting after the second injection. Peak anti-*Na*-GST-1 IgG levels were observed between 2 and 4 weeks following the third dose, regardless of *Na*-GST-1 formulation. IgG levels decreased but remained significantly above baseline in all groups by day 290, at which point all participants (20 of 20 evaluable participants) still had detectable IgG. Longitudinal antigen-specific IgG1 and IgG3 subclass responses mirrored those of total IgG, whereas IgG4 responses were lower in the groups that received the vaccine with the CpG adjuvant compared to the non-CpG group.

## Conclusions

Vaccination of hookworm-naïve adults with *Na*-GST-1/Alhydrogel plus CpG 10104 was safe and minimally reactogenic. Addition of CpG 10104 to *Na*-GST-1/Alhydrogel resulted in significant improvement in IgG responses against the vaccine antigen. These promising results have led to inclusion of the CpG 10104 formulation of *Na*-GST-1/Alhydrogel in a Phase 2 proof-of-concept controlled human infection trial.

### Author summary

Infection caused by *Necator americanus* is a major neglected tropical disease with significant associated morbidity. New tools, including vaccines, are needed due to the inadequacy of current control strategies. *N. americanus* Glutathione-S-Transferase-1 (*Na*-GST-1) is a leading hookworm vaccine candidate. Antibodies induced by this vaccine are postulated to interfere with the host blood digestion pathway of adult *N. americanus* worms, thereby impairing their development and survival. We conducted a Phase 1 trial of recombinant *Na*-GST-1 adjuvanted with Alhydrogel in 24 healthy, hookworm-naïve adults living in the metropolitan Washington DC region. Each participant received three intramuscular injections every 2 months of the vaccine administered with or without a Cytosine-phospho-Guanine (CpG) oligodeoxynucleotide (ODN) Toll-like receptor-9 agonist, CpG 10104. *Na*-GST-1/Alhydrogel was well tolerated in this study, and no vaccine-related serious adverse events were observed. Significant levels of antigen-specific IgG antibodies were induced with the highest responses seen 2 to 4 weeks after the third vaccination. The addition of CpG 10104 to the vaccine resulted in significant improvement in the antigen-specific IgG responses. Based on these results, the vaccine has been advanced into a Phase 1 clinical trial in a hookworm-endemic region of Africa as well as a Phase 2 controlled human infection study to test proof of efficacy.

## Introduction

Hookworm is a leading cause of anemia in children and women of childbearing age in tropical regions of the world, with over 400 million people estimated to be infected [1]. An effective hookworm vaccine would prevent this burden of disease and overcome the significant shortcomings of the current disease control strategy based on periodic mass drug administration of a benzimidazole such as albendazole.

*Necator americanus* Glutathione-S-Transferase-1 (*Na*-GST-1) is one of two leading hookworm vaccine candidate antigens currently in clinical development for the most prevalent hookworm species, accounting for an estimated 80% of infections worldwide [2]. It, together with the aspartic protease *Na*-APR-1, is a critical component of the parasite's blood-feeding pathway. To date, a series of Phase 1 trials of recombinant *Na*-GST-1 have shown the vaccine to be safe and well tolerated in hookworm-naïve and hookworm-exposed adults as well as children [3–5]. These trials have tested different adjuvant formulations of the recombinant protein, including Alhydrogel and Alhydrogel plus an aqueous formulation of Glucopyranosyl Lipid Adjuvant (GLA-AF).

Although it is assumed that induction of antibodies to *Na*-GST-1 will be essential to this vaccine's efficacy, the level of antibodies necessary to achieve protection is unknown. The hypothesized mechanism of protection of the vaccine is via induction of anti-*Na*-GST-1 antibodies that are ingested by developing hookworms after host infection. In the digestive tract of the hookworm, these antibodies will bind to the native protein and block its ability to detoxify free heme, a byproduct of the worm's blood digestion pathway, leading to impaired growth and death of the worm. Given this putative mechanism of vaccine-induced protection, it is likely that production of higher levels of antibodies will result in improved efficacy.

Cytosine-phospho-Guanine (CpG) oligodeoxynucleotides (ODN) are Toll-like receptor (TLR)-9 agonists have been used as adjuvants in approved vaccines to improve protective antibody levels and seroresponse rates for hepatitis B [6], COVID-19 [7], and most recently, anthrax [8]. We hypothesized that the addition of a CpG ODN adjuvant to the Alhydrogel formulation of recombinant *Na*-GST-1 would result in increased antigen-specific IgG responses compared to the Alhydrogel formulation without CpG, thereby increasing the likelihood of protective efficacy of this product.

## Materials and methods

### Ethics statement

The study was approved by the George Washington University (GW) Institutional Review Board (IRB), approval number 041436. Written informed consent was documented for each study participant using an IRB-approved consent form prior to initiation of any study-related procedures. The trial was conducted after submission of an investigational new drug application (number 016156) to the US Food and Drug Administration and is registered at *clinicaltrials.gov* (NCT02143518).

### Study vaccines

Recombinant *Na*-GST-1 was manufactured as reported previously [9]. It was supplied to the study site in 2.0 mL vials containing 1.35 mL of a 0.1 mg/mL suspension of *Na*-GST-1 absorbed to 0.8 mg/mL Alhydrogel (Croda, Denmark) in a buffer of 10% glucose and 10 mM imidazole (pH 7.4). *Na*-GST-1/Alhydrogel was manufactured, formulated, and vialed at Aeras Global Vaccine Foundation (Rockville, Maryland, USA).

CpG 10104 is a short synthetic ODN (5'-TCG TCG TTT CGT CGT TTT GTC GTT-3') that was supplied as an aqueous solution in multi-dose vials at a concentration of 2.0 mg/mL.

Appropriate volumes of the CpG 10104 solution were added to vials containing *Na*-GST-1/ Alhydrogel between 30 minutes and 4 hours of administration for all dosage groups to ensure an amount of 500 μg ODN per dose. CpG 10104 was manufactured according to current Good Manufacturing Practice by the Access to Advanced Health Institute (AAHI, Seattle, WA).

A toxicology study conducted in Sprague-Dawley rats according to current Good Laboratory Practice was completed prior to initiation of the trial reported herein, in which there were no observed adverse effects of recombinant *Na*-GST-1 adjuvanted with Alhydrogel with or without the addition of CpG 10104 on survival, body weights, or organ weights or any clinical signs of toxicity or abnormal gross necropsy findings.

## Study site and population

This was a randomized, observer-blind, dose-escalation Phase 1 clinical trial in healthy, hookworm-naïve adults conducted from November 2014 to October 2016 at GW in Washington, District of Columbia. The primary objective was to estimate the frequency of vaccine-related adverse events for *Na*-GST-1/Alhydrogel administered alone or in combination with CpG 10104. Determination of the antigen dose and formulation that generated the highest anti-*Na*-GST-1 IgG antibody response and assessment of the duration of the IgG response were secondary objectives.

Inclusion criteria included being male or a non-pregnant female between 18 and 50 years of age and in good general health as determined by screening procedures. Exclusion criteria included being pregnant or breast-feeding; having evidence of clinically significant systemic disease or a major psychiatric condition; history of severe allergic reaction or anaphylaxis; severe asthma; chronic hepatitis B, C, or HIV infection; receipt of corticosteroids or other immunosuppressive drugs; known immunodeficiency; receipt of a live vaccine within 4 weeks or an inactivated vaccine within 2 weeks of enrollment; receipt of blood products within the previous 6 months; pre-existing autoimmune or antibody-mediated diseases; and history of previous hookworm infection or residence for more than 6 months in a hookworm-endemic area.

## Clinical procedures

Twenty-four adults were progressively enrolled into two cohorts. In Cohort 1, volunteers were randomized to receive 30 μg *Na*-GST-1/Alhydrogel co-administered with 500 μg CpG 10104 (n = 8) or 100 μg *Na*-GST-1/Alhydrogel (n = 4). In Cohort 2, volunteers were randomized to receive 100 μg *Na*-GST-1/Alhydrogel co-administered with 500 μg CpG 10104 (n = 8) or 100 μg *Na*-GST-1/Alhydrogel (n = 4).

Immunizations were delivered by intramuscular (IM) injection in the deltoid muscle, with successive vaccinations given in alternating arms. Three injections of study vaccine were administered to each participant on approximately study days 0, 56, and 112. Participants were directly observed for immediate reactions for at least 1 hour after immunization, followed by clinical assessments at the study clinic on study days 3, 7, 14, and 28 following vaccination, and 3 and 6 months after the third vaccination. Telephone safety calls were conducted 9 and 12 months after the third vaccination. The two cohorts were enrolled in a staggered fashion so that safety and reactogenicity data for the 14 days following administration for Cohort 1 were reviewed by the Medical Monitor and the Safety Monitoring Committee (SMC) prior to initiating vaccinations for Cohort 2.

## Blinding

Participants, investigators, clinical staff conducting study-related assessments, and laboratory personnel performing immunology assays were blinded to study product allocation. Within

each cohort, allocation to *Na*-GST-1/Alhydrogel or *Na*-GST-1/Alhydrogel/CpG 10104 was performed using a computer-generated randomization code provided by the Sponsor's data management contractor to the study investigational pharmacist, who was the only study team member aware of participant vaccine allocation. The study pharmacist prepared all vaccine doses in a separate room and the vaccine-filled syringes were delivered to the vaccinators with the contents of all syringes disguised with opaque tape.

## Assessment of safety and tolerability

The incidence of solicited adverse events (AEs) was monitored for 14 days following each vaccination. Solicited injection site AEs included pain, tenderness, erythema, and swelling, whereas solicited systemic AEs were nausea, vomiting, headache, myalgia, arthralgia, urticaria, rash, and fever. Unsolicited AEs were evaluated up until 6 months following the third vaccination. In addition, adverse events of special interest (AESIs), medically attended adverse events (MAAEs), and serious adverse events (SAEs) were evaluated for the duration of study participation. Due to the inclusion of the novel CpG 10104 adjuvant in the study, active surveillance for the following AESIs was performed: autoimmune musculoskeletal disorders (e.g., systemic lupus erythematosus, Sjögren's syndrome, rheumatoid arthritis), neuroinflammatory disorders (e.g., multiple sclerosis, optic neuritis, Guillain-Barré syndrome), metabolic diseases (e.g., autoimmune thyroiditis), gastrointestinal disorders (e.g., inflammatory bowel disease), vasculitides, and other autoimmune or inflammatory diseases [10].

AE severity was assessed using the following grading scheme: Grade 1 (mild), no effect on activities of daily living and no medical intervention or therapy required; Grade 2 (moderate), partial limitation in activities of daily living (can complete >50% of baseline) and no or minimal medical intervention or therapy required; Grade 3 (severe), activities of daily living limited to <50% of baseline and medical evaluation or therapy required; and, Grade 4 (potentially life threatening). All AEs were assessed by a blinded study investigator as being definitely, probably, possibly, unlikely, or not related to study vaccine. Injection site swelling and erythema were assessed as mild (25 to 50 mm in diameter), moderate (51 to 100 mm), or severe (>100 mm); whereas oral temperature was graded as mild (38.0 °C to 38.4 °C), moderate (38.5 °C to 38.9 °C), or severe (≥39.0 °C).

Peripheral blood samples were obtained for hematology and biochemistry clinical laboratory evaluations immediately before each vaccination, plus 14 days post-vaccination. Antinuclear antibody (ANA), rheumatoid factor, and anti-double-stranded DNA (anti-dsDNA) antibody tests were also performed on these study days and on study day 270, due to the inclusion of CpG 10104 as an immunostimulant. Clinical laboratory AEs were assessed using a standardized toxicity table for hemoglobin, platelets, white blood cell count (WBCs), absolute neutrophil count (ANC), serum creatinine, and alanine aminotransferase (ALT) [11].

## Immunology methods

Antigen-specific immunoglobulin G (IgG) and IgG subclass (IgG1, IgG3, and IgG4) antibody levels to *Na*-GST-1 were measured in serum samples using qualified indirect ELISAs, as described previously [3,12], on each day of vaccination, 14 and 28 days after each vaccination, and 3 and 6 months after the third vaccination. Briefly, 96-well microwell plates (Nunc Maxisorb) were adsorbed with recombinant *Na*-GST-1, incubated with test sera, blocked with buffer consisting of 5% Bovine Serum Albumin in 1X phosphate-buffered saline with Tween 20 (PBST20), and stored overnight at 4–8 °C. Plates were washed, and a 1:1000 dilution in PBST20 of the following horseradish peroxidase-conjugated anti-human secondary antibodies

were added: IgG (Invitrogen JDC-10), anti-human IgG1 (Invitrogen HP6070), anti-human IgG3 (Southern Biotech HP6050), and anti-human IgG4 (Invitrogen HP6025). Plates were then covered and stored at ambient temperature for 120 minutes. Plates were rewashed and a chromogenic substrate was added, with the Optical Density (OD) of the samples read at 492 nm ($OD_{492}$) by a SpectraMax 348 PC 340 ELISA plate reader (Molecular Devices, Sunnyvale, CA), with the data captured and interpolated by SoftMax Pro GxP 5.4 (Molecular Devices, Sunnyvale, CA).

Levels of IgG antibodies against *Na*-GST-1 were converted to Arbitrary Units (AU) by homologous interpolation of $OD_{492}$ readings from a standard calibration curve derived from serial dilutions of a human standard reference serum. A reactivity threshold (RT) of 2.49 AU for the anti-*Na*-GST-1 IgG assay was calculated from a standard calibration curve as described previously [12]. Levels of IgG subclasses (IgG1, IgG3, and IgG4) against *Na*-GST-1 in sera were converted to AUs by heterologous interpolation of their $OD_{492}$ readings onto a standard calibration curve derived from serial dilutions of an appropriate human myeloma-derived IgG1, IgG3, or IgG4 purified antibody. A higher (2-fold) serum dilution was used to determine the levels of anti-*Na*-GST-1 IgG4 in test samples that resulted when the $OD_{492}$ absorbance levels were saturated at a lower dilution.

## Statistical methods

As a Phase 1 trial, this study was not powered to detect statistically significant differences between groups. The sample size of 24 participants is within the range frequently used in Phase 1 trials for the initial assessment of the safety, tolerance, and immunogenicity of an investigational vaccine. All participants who received at least one dose of vaccine were included in the safety analyses whereas immunogenicity data from participants who did not receive all three planned injections were included in the analyses only up until the study day of the dose that was not administered.

AEs were categorized according to the Medical Dictionary of Regulatory Activities (MedDRA, Version 15.0) System, Organ, and Class (SOC) and preferred terms. The number and percentage of study participants experiencing each AE were tabulated by vaccine group. Data from participants that received 100 μg *Na*-GST-1/Alhydrogel in the two cohorts were pooled. Summary statistics (e.g., mean, median, minimum, and maximum) were calculated for each hematology, clinical chemistry, and autoantibody parameter. Formal statistical comparisons were not conducted given the small group sizes.

Geometric mean antibody levels (GMLs) were calculated for each vaccine formulation and dose group, with 95% confidence intervals (CIs) estimated using a t-distribution and the geometric standard deviation in each group. Comparisons of antibody levels between groups were made by one-way analysis of variance (ANOVA) tests with pair-wise comparisons between vaccine doses and formulations made by Fisher's Exact tests. Participants were considered seropositive when an anti-*Na*-GST-1 IgG level was above the RT (2.49 AU) for the ELISA assay.

## Results

### Participant flow and baseline data

A total of 45 adults initiated the screening process with a final enrollment of 24 participants (see Fig 1 for participant flow). Of the 21 adults who were not enrolled, 4 were lost to follow-up during the screening process; 9 had exclusionary medical or psychiatric conditions; 7 had exclusionary abnormal screening laboratory test results; and 1 was already participating in another trial of an investigational product. Demographic and baseline characteristics of enrolled participants are summarized in Table 1.

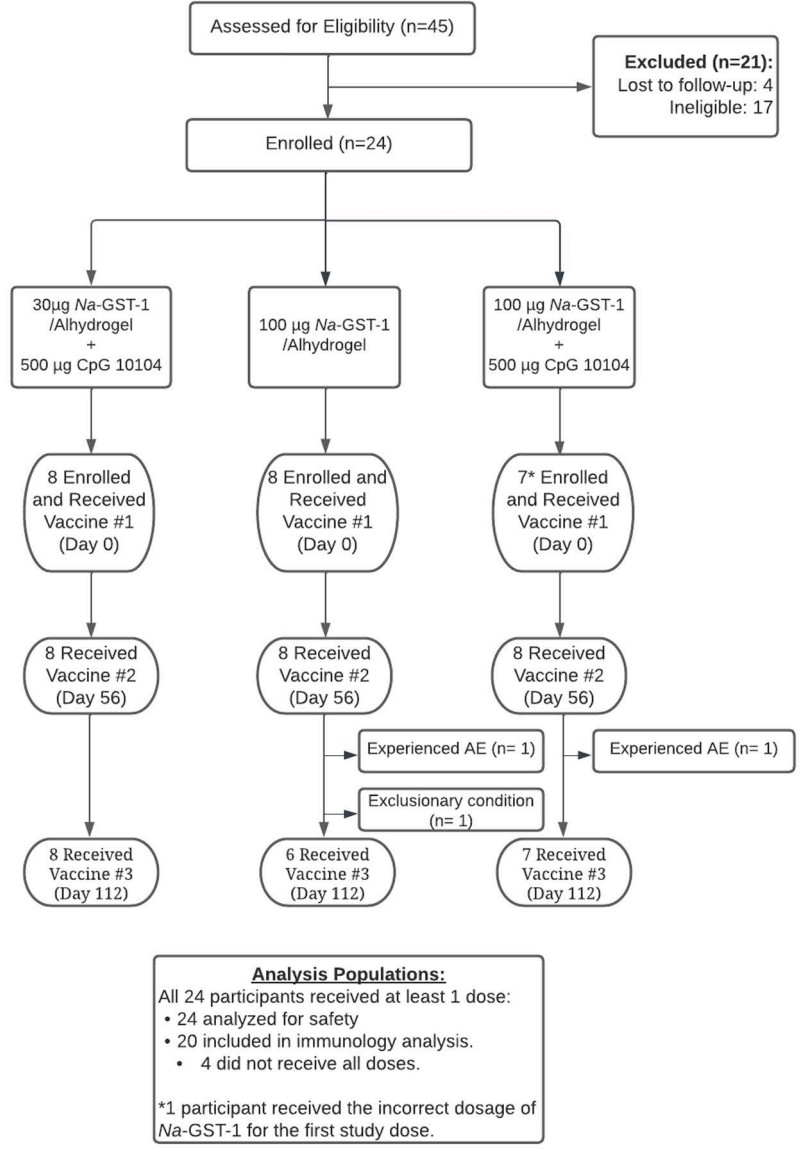

**Fig 1. Participant flow (CONSORT diagram).**

All 24 enrolled participants received at least 1 dose of the study vaccine, while 20 (83.3%) participants received all 3 planned doses. Two participants in the 100 μg *Na*-GST-1/Alhydrogel group did not receive the third dose: one participant developed an elevated ANA following the second injection and the other participant did not receive the third and final injection due to the discovery of previously undisclosed past history of an exclusionary health condition (hypoparathyroidism). Two participants in the 100 μg *Na*-GST-1/Alhydrogel + 500 μg CpG 10104 group did not receive all doses, with one participant receiving the incorrect dose (30 μg *Na*-GST-1/Alhydrogel + 500 μg CpG 10104) for the first injection, while one participant did not receive the third and final injection due to mild-to-moderate reactogenicity (injection site pain and tenderness, headache, myalgia, arthralgia, fatigue, and fever) following the first two injections plus unavailability to attend future study visits due to a new job. Two additional participants in the 30 μg *Na*-GST-1/Alhydrogel + 500 μg CpG 10104 group

**Table 1. Demographic and baseline characteristics of study participants by vaccine group.**

|  | 30 μg *Na*-GST-1/ Alhydrogel + 500 μg CpG | 100 μg *Na*-GST-1/ Alhydrogel | 100 μg *Na*-GST-1/ Alhydrogel + 500 μg CpG | Overall |
|---|---|---|---|---|
|  | (N = 8) | (N = 8) | (N = 8) | (N = 24) |
| **Age (years)** |  |  |  |  |
| Mean (range) | 34.4 (21–50) | 29.4 (20–49) | 34.4 (21–49) | 32.7 (20–50) |
| **Gender (%)** |  |  |  |  |
| Female | 2 (25.0%) | 5 (62.5%) | 3 (37.5%) | 10 (41.7%) |
| Male | 6 (75%) | 3 (37.5%) | 5 (62.5%) | 14 (58.3%) |
| **Race (%)** |  |  |  |  |
| American Indian/ Alaskan Native | 0 (0.0%) | 0 (0.0%) | 0 (0.0%) | 0 (0.0%) |
| Asian | 0 (0.0%) | 2 (25.0%) | 0 (0.0%) | 2 (8.3%) |
| Black/African American | 5 (62.5%) | 1 (12.5%) | 4 (50%) | 10 (41.7%) |
| Native Hawaiian/ Pacific Islander | 1 (12.5%) | 0 (0.0%) | 0 (0.0%) | 1 (4.2%) |
| White | 1 (12.5%) | 5 (62.5%) | 4 (50%) | 10 (41.7%) |
| Other | 1 (12.5%) | 0 (0.0%) | 0 (0.0%) | 1 (4.2%) |

were lost to follow-up after the final in-person clinic visit but before completion of the final two safety telephone checks.

## Safety results

One participant was discontinued from the study due to an AE, based on the recommendation of the study SMC after review of blinded safety data. This participant had an increase in ANA titer following the second dose of vaccine. The recommendation to discontinue further vaccinations, made without unblinding the study participant, was due to a hypothetical concern that CpG 10104 might induce production of anti-DNA antibodies. However, unblinding after database lock revealed that this participant had received 100 μg *Na*-GST-1/Alhydrogel *without* CpG 10104.

**Solicited AEs.** Of the 24 participants, 4 reported a total of 9 solicited immediate injection site AEs (pain and/or tenderness) within 1 hour post-injection. Seven of 9 immediate AEs were mild in severity, while 2 were moderate. Seven of the immediate AEs resolved within 4 days of onset, whereas 1 participant who received with 30 μg *Na*-GST-1/Alhydrogel + 500 μg CpG had immediate mild injection site pain and tenderness that lasted 13 and 31 days after the first vaccination, respectively. Seven of the reported immediate AEs were reported by participants in the 30 μg *Na*-GST-1/Alhydrogel + 500 μg CpG group and 2 occurred in participants who received 100 μg *Na*-GST-1/Alhydrogel. No participants in the 100 μg *Na*-GST-1/Alhydrogel + 500 μg CpG group reported any solicited immediate AEs.

As shown in Table 2, the most common solicited injection site AEs were pain and tenderness. Two severe injection site reactions were reported: 1 occurrence of severe injection site tenderness, and 1 of severe injection site swelling. For each of these participants, the severe injection site reactions were reported for only 1 of the 3 injections. These 2 participants were in different vaccine groups: 1 received 100 μg *Na*-GST-1/Alhydrogel without CpG 10104, while 1 received 100 μg *Na*-GST-1/Alhydrogel + 500 μg CpG 10104. Statistical comparison of injection site reaction frequencies did not reveal any significant differences either in incidence or intensity of injection site AEs between the three vaccine groups, or by injection number

**Table 2. Number and percentage of participants experiencing solicited injection site and systemic reactions by maximum severity and vaccine group.**

| | 30 µg *Na*-GST-1/Alhydrogel + 500 µg CpG 10104 | | | 100 µg *Na*-GST-1/Alhydrogel | | | 100 µg *Na*-GST-1/Alhydrogel + 500 µg CpG 10104 | | |
| --- | --- | --- | --- | --- | --- | --- | --- | --- | --- |
| | (N = 8) | | | (N = 8) | | | (N = 8) | | |
| | Mild | Moderate | Severe | Mild | Moderate | Severe | Mild | Moderate | Severe |
| *Injection site events* | | | | | | | | | |
| Pain | 7 (87.5%) | 2 (25.0%) | – | 6 (75.0%) | 1 (12.5%) | – | 6 (75.0%) | 3 (37.5%) | – |
| Tenderness | 6 (75.0%) | 4 (50.0%) | – | 6 (75.0%) | 4 (50.0%) | – | 5 (62.5%) | 5 (62.5%) | 1 (12.5%) |
| Erythema | 1 (12.5%) | – | – | – | 1 (12.5%) | – | – | – | – |
| Swelling | 1 (12.5%) | – | – | – | – | 1 (12.5%) | – | 1 (12.5%) | – |
| *Systemic events* | | | | | | | | | |
| Nausea | – | 1 (12.5%) | – | 1 (12.5%) | – | – | 3 (37.5%) | 1 (12.5%) | – |
| Vomiting | – | 1 (12.5%) | – | 1 (12.5%) | – | – | 1 (12.5%) | – | – |
| Headache | 3 (37.5%) | 1 (12.5%) | – | 2 (25.0%) | – | – | 2 (25.0%) | 2 (25.0%) | – |
| Myalgia | – | – | – | – | – | – | 1 (12.5%) | 4 (50.0%) | – |
| Arthralgia | 1 (12.5%) | – | – | – | – | – | 4 (50.0%) | – | – |
| Fever | – | – | – | 1 (12.5%) | – | – | 1 (12.5%) | 1 (12.5%) | – |

Note: N = Number of participants in the safety population who received the specified vaccine. A participant was counted only once per severity grade.

(i.e., after the first, second, or third injections within vaccine groups). Similarly, the incidence of injection site AEs was not significantly different between those who received *Na*-GST-1/Alhydrogel compared to those who received *Na*-GST-1/Alhydrogel in combination with CpG 10104 (pooling the 30 µg and 100 µg *Na*-GST-1/Alhydrogel groups).

The most frequently observed solicited systemic AEs across all vaccine groups was headache. Most solicited systemic AEs reported by participants were mild in severity, and no severe systemic reactions were recorded in any participant. Systemic reactions were observed more frequently in the 100 µg *Na*-GST-1/Alhydrogel + 500 µg CpG group than in the other 2 groups, with 37.5%, 25%, and 42.9% of participants experiencing at least one symptom after the first, second, and third injections, respectively.

**Unsolicited AEs.** A total of 203 unsolicited AEs were observed in study participants, with 35 (17.2%) assessed as definitely, probably, or possibly related to the study vaccine (S1 Table). All 24 participants experienced at least one unsolicited AE, with the most common events falling under the 'Infections and Infestations' MedDRA SOC, in 19 of 24 (79.2%) participants. The majority of these were upper respiratory infections, all of which were deemed unrelated to vaccination. Related unsolicited AEs occurred in approximately equal proportions of participants in the 3 vaccine groups, although a greater absolute number of events occurred in those receiving the 100 µg dose of *Na*-GST-1 with 14 events in the 100 µg *Na*-GST-1/Alhydrogel group and 12 events in the 100 µg *Na*-GST-1/Alhydrogel plus CpG 10104 groups compared to 9 events in the 30 µg *Na*-GST-1/Alhydrogel plus CpG 10104 group. The most common vaccine-related unsolicited AEs were mild, transient clinical laboratory abnormalities (detailed below) and headache outside of the reactogenicity period.

**Clinical laboratory AEs.** Overall, 20 clinical laboratory AEs were observed in 9 (37.5%) of the 24 participants (S2 Table). Mild decreases in WBC count were observed in 7 (29.2%) participants, of which 3 were possibly related (due to timing post-vaccination and lack of alternative etiology) and 4 were unlikely related to study vaccine. There were 4 mild decreases in ANC count in 2 participants (8.3%), both in the group that received 100 µg *Na*-GST-1/Alhydrogel plus CpG 10104. Mildly decreased hemoglobin levels were observed in 4 of 24 participants (16.7%). Two (8.3%) participants, 1 in each of the 30 µg and 100 µg *Na*-GST-1/

Alhydrogel plus CpG 10104 groups, experienced mild, transient, asymptomatic increases in ALT. No vaccine related abnormal chemistry results or increases in creatinine concentration were reported for any group.

All clinical laboratory AEs were mild in severity. No concerning differences between vaccine groups and/or CpG status were identified. Mild transient decreases in WBC and ANC counts after vaccination were slightly more frequent in participants who received 100 µg *Na*-GST-1/Alhydrogel plus CpG 10104 compared to the other groups, although the overall number of events was low.

## Anti-*Na*-GST-1 IgG responses

Table 3 summarizes the anti-*Na*-GST-1 IgG levels of each vaccine group by study day, while Fig 2 shows the geometric mean longitudinal IgG responses. The highest anti-*Na*-GST-1 IgG GMLs were observed in participants that received 100 µg *Na*-GST-1/Alhydrogel plus CpG 10104, with a peak of 392.0 AU on study day 140, 4 weeks after the final vaccination. Although the GMLs were not significantly different between the 30 µg and 100 µg *Na*-GST-1/Alhydrogel plus CpG 10104 groups ($P = 0.613$) at any timepoint, they were both significantly higher on study day 126 compared to the 100 µg *Na*-GST-1/Alhydrogel recipients ($P = 0.038$ for both pairwise comparisons).

Baseline anti-*Na*-GST-1 IgG levels were undetectable or only slightly higher than the RT in all study participants (Table 3), but they progressively increased in all groups after each injection, peaking on study day 140. IgG levels subsequently decreased slowly in all three vaccine groups but remained significantly higher than baseline at the final study visit on Day 290 (six months after final vaccination), particularly in the 100 µg *Na*-GST-1/Alhydrogel plus CpG 10104 group. As shown in Table 3, significant differences in IgG responses between the three vaccine groups first appeared on study Day 70, 2 weeks after the second doses of vaccine were administered ($P = 0.001$, overall ANOVA), with pair-wise comparisons demonstrating that both the 30 µg and 100 µg *Na*-GST-1/Alhydrogel plus CpG 10104 groups had higher IgG levels than the 100 µg *Na*-GST-1/Alhydrogel group, without significant differences between the 30 µg and 100 µg *Na*-GST-1/Alhydrogel plus CpG 10104 groups. This pattern remained for all subsequent time points except on study Day 200 (3 months after final vaccinations), when the overall ANOVA was not statistically significant ($P = 0.098$).

Table 3 also summarizes the proportions of anti-*Na*-GST-1 IgG seroresponders at each time point for each vaccine group. Response rates to vaccination were uniformly high across all three vaccine groups: starting on study day 70 (2 weeks post-second vaccination), 100% of participants in all groups were seropositive, which was maintained for the duration of the study. No significant differences in response rates were observed between vaccine groups at any study time point.

## IgG subclass responses

Antigen-specific IgG1 and IgG3 subclass responses followed similar patterns over time for all vaccine groups, although for IgG1, the highest responses were in the 100 µg *Na*-GST-1/Alhydrogel plus CpG 10104 group whereas for IgG3, the highest responses were in the 30 µg *Na*-GST-1/Alhydrogel plus CpG 10104 group (Fig 3). Responses for these IgG subclasses were higher for both of the *Na*-GST-1 doses in the CpG 10104 formulation compared to the non-CpG 100 µg antigen dose group. In contrast to the antigen-specific IgG1 and IgG3 responses, the highest anti-*Na*-GST-1 IgG4 levels were seen in the non-CpG 100 µg *Na*-GST-1/Alhydrogel group. In addition, IgG4 responses were minimal in all groups until after the third vaccination, after which the levels continuously increased until the end of the study on day 290.

**Table 3.  Anti-*Na*-GST-1 IgG antibody levels and seroresponse rates by vaccine group and study day.**

| Time point | 30 µg *Na*-GST-1/Alhydrogel + 500 µg CpG | 100 µg *Na*-GST-1/ Alhydrogel | 100 µg *Na*-GST-1/Alhy-drogel + 500 µg CpG |
|---|---|---|---|
| **Day 0** (Pre-Dose 1) | | | |
| N | 8 | 8 | 8 |
| GML (95% CI) | 3.02 (2.07–4.41) | 3.34 (2.25–4.97) | 2.49 (2.49–2.49) |
| Number responders (%) | 1 (12.5%) | 2 (25.0%) | 0 (0.0%) |
| **Day 14** | | | |
| N | 8 | 8 | 8 |
| GML (95% CI) | 3.38 (2.24–5.13) | 4.48 (2.82–7.12) | 4.23 (2.46–7.26) |
| Number responders (%) | 2 (25.0%) | 4 (50.0%) | 3 (37.5%) |
| **Day 28** | | | |
| N | 8 | 8 | 8 |
| GML (95% CI) | 14.92 (7.58–29.36) | 4.73 (2.85–7.85) | 9.7 (3.96–23.74) |
| Number responders (%) | 8 (100.0%) | 4 (50.0%) | 5 (62.5%) |
| **Day 56** (Pre-Dose 2) | | | |
| N | 8 | 8 | 8 |
| GML (95% CI) | 6.83 (3.64–12.79) | 4.89 (3.21–7.45) | 7.49 (3.85–14.56) |
| Number responders (%) | 5 (62.5%) | 5 (62.5%) | 5 (62.5%) |
| **Day 70** (14 Days Post-Dose 2) | | | |
| N | 8 | 8 | 8 |
| GML (95% CI) | 321.1[†] (164.6–626.3) | 16.6 (11.23–24.54) | 173.0[†] (82.19–364.0) |
| Number responders (%) | 8 (100.0%) | 8 (100.0%) | 8 (100.0%) |
| **Day 84** (28 Days Post-Dose 2) | | | |
| N | 8 | 8 | 8 |
| GML (95% CI) | 157.7[†] (85.93–289.5) | 15.86 (10.71–23.47) | 208.3[†] (68.87–630.2) |
| Number responders (%) | 8 (100.0%) | 8 (100.0%) | 8 (100.0%) |
| **Day 112** (Pre-Dose 3) | | | |
| N | 8 | 8 | 8 |
| GML (95% CI) | 108.5[†] (61.9–190.1) | 12.72 (9.0–17.99) | 63.3[†] (33.98–117.9) |
| Number responders (%) | 8 (100.0%) | 8 (100.0%) | 8 (100.0%) |
| **Day 126** (14 Days Post-Dose 3) | | | |
| N | 8 | 6 | 7 |
| GML (95% CI) | 300.7[*] (194.6–464.8) | 49.42 (17.07–143.0) | 389.5[*] (212.8–713.0) |
| Number responders (%) | 8 (100.0%) | 6 (100.0%) | 7 (100.0%) |
| **Day 140** (28 Days Post-Dose 3) | | | |
| N | 8 | 6 | 7 |
| GML (95% CI) | 308.8 (139.0–685.8) | 57.1 (20.12–162.0) | 392.0[*] (224.3–684.9) |
| Number responders (%) | 8 (100.0%) | 6 (100.0%) | 7 (100.0%) |
| **Day 200** | | | |
| N | 8 | 6 | 7 |
| GML (95% CI) | 85.18 (49.56–146.4) | 30.58 (14.55–64.27) | 113.5 (64.38–200.1) |
| Number responders (%) | 8 (100.0%) | 6 (100.0%) | 7 (100.0%) |
| **Day 290** (Follow-up) | | | |
| N | 8 | 5 | 7 |
| GML (95% CI) | 45.32 (27.75–74.01) | 17.51 (10.73–28.56) | 67.36 (28.85–157.3) |
| Number responders (%) | 8 (100.0%) | 5 (100.0%) | 7 (100.0%) |

*Na*-GST-1 = *N. americanus* glutathione S-transferase-1. GML = geometric mean level. [*]*P* < 0.05 for comparison with 100 µg *Na*-GST-1/Alhydrogel comparator group (Fisher's exact test). [†]*P* < 0.01 for comparison with 100 µg *Na*-GST-1/Alhydrogel comparator group (Fisher's exact test). Seropositivity was defined as having an antibody level about the reactivity threshold for the assay.

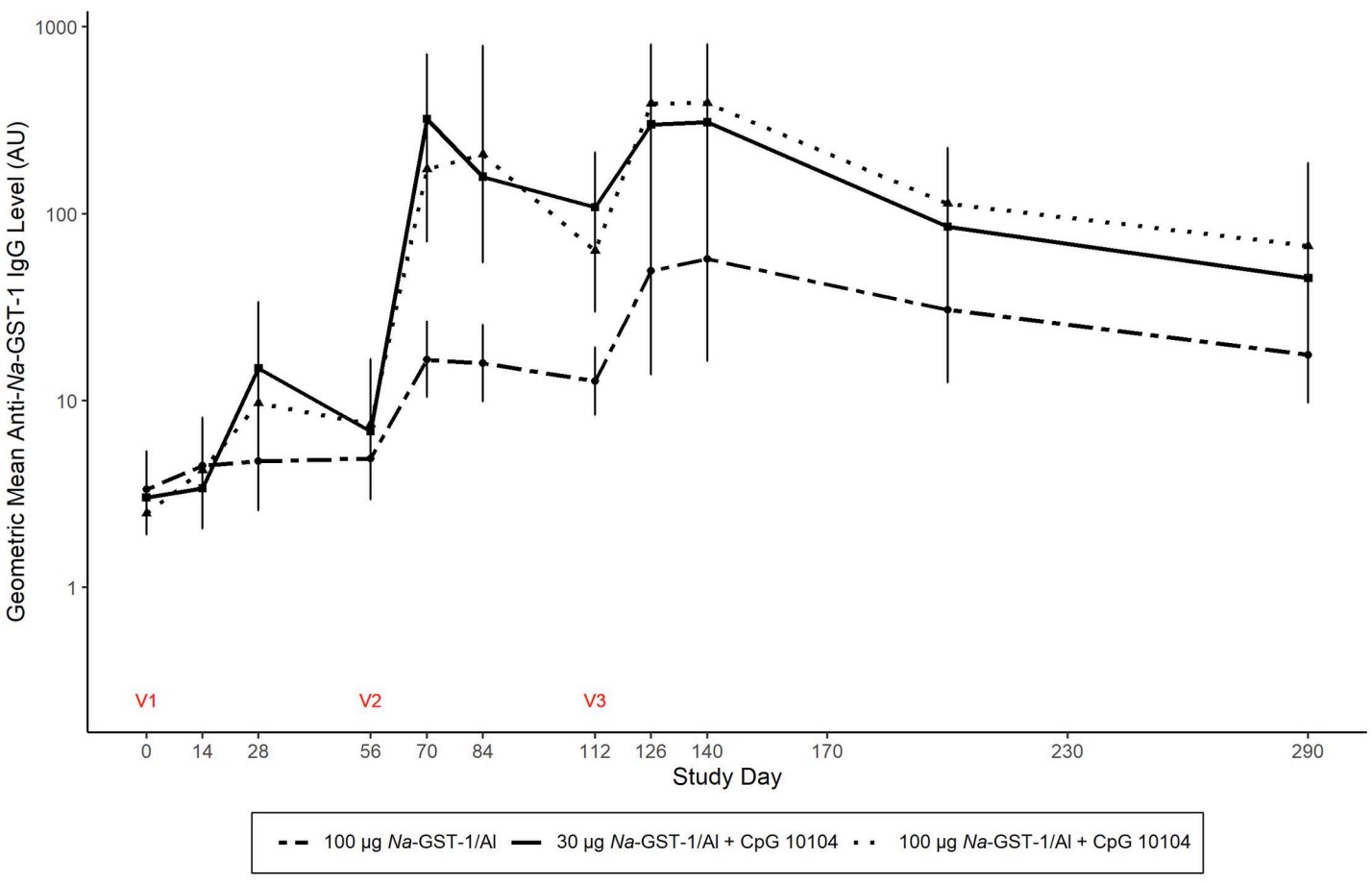

**Fig 2.  Geometric mean anti-*Na*-GST-1 IgG levels over time by vaccine group, as measured by ELISA (Arbitrary Units).** Per-protocol immunogenicity population. Vaccinations were administered on study days 0, 56, and 112. Error bars represent 95% confidence intervals. Recipients of 100 µg *Na*-GST-1/Alhydrogel (without CpG 10104) were pooled across cohorts.

## Discussion

In this Phase 1, first-in-humans trial of the *Na*-GST-1/Alhydrogel vaccine co-administered with CpG 10104, we demonstrated significant boosting of antigen-specific IgG responses with the addition of the ODN immunostimulant. Furthermore, the vaccine was safe and well tolerated in healthy, hookworm-naïve adult volunteers, and the addition of CpG 10104 to the Alhydrogel formulation of recombinant *Na*-GST-1 did not substantially increase either the frequency or severity of solicited or unsolicited AEs. However, given the relatively small group sizes, the safety of this vaccine formulation will need to be further evaluated in future studies.

CpG ODNs are short, single-stranded, synthetic fragments of DNA that contain unmethylated CpG dinucleotide motifs commonly found in most bacterial genomes [13]. CpG ODNs are ligands for TLR9 that are expressed by cells of the immune system such as plasmacytoid dendritic cells and B cells. After binding to TLR9, CpG ODNs activate the cells to produce proinflammatory cytokines and activate natural killer cells, neutrophils, and monocytes, eventually leading to activation of Th1 immune responses. When added to a recombinant protein-based vaccine, such activation is intended to improve IgG antibody responses to the vaccine antigen, thereby increasing the potential vaccine efficacy in preventing the infection in question.

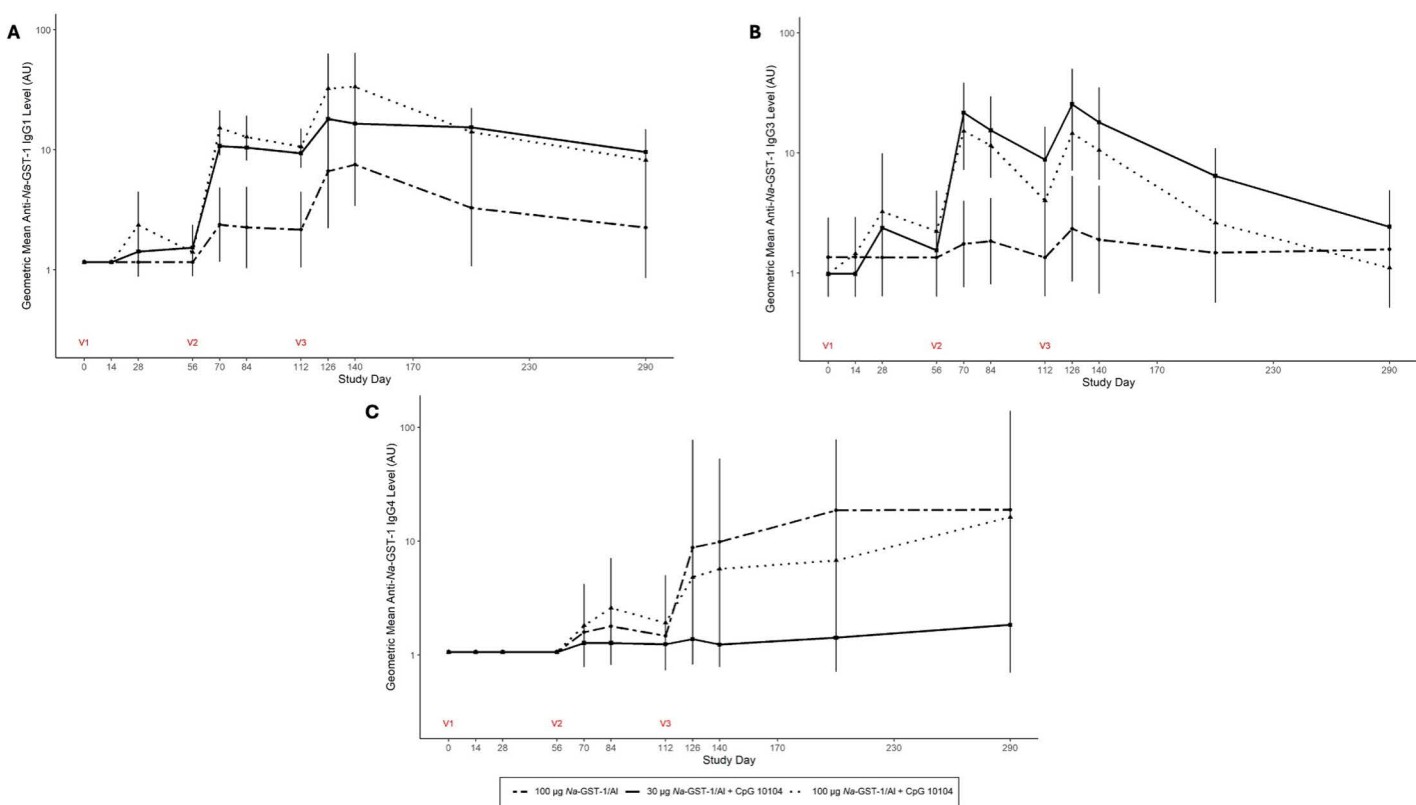

**Fig 3. Geometric mean anti-*Na*-GST-1 IgG subclass levels over time.** Subclass levels as measured by ELISA (Arbitrary Units) are shown by vaccine group for: (A) IgG1, (B) IgG3, and (C) IgG4. Per-protocol immunogenicity population. Vaccinations were administered on study days 0, 56, and 112. Error bars represent 95% confidence intervals. Recipients of 100 μg *Na*-GST-1/Alhydrogel (without CpG 10104) were pooled across cohorts.

Synthetic CpG ODNs have successfully been used to improve antibody responses and rates of seroconversion for a number of investigational and licensed vaccines, including the CYFENDUS anthrax vaccine recently approved by the FDA [8], the Heplisav-B vaccine that is approved to protect against Hepatitis B virus, and the Corbevax COVID-19 vaccine that has been authorized by the World Health Organisation for emergency use listing, was approved in India and Indonesia, and administered to over 100 million adults and children in Asia [14]. The latter two vaccines both contain CpG 1018, a B-class CpG ODN that is similar in sequence to the CpG 10104 adjuvant tested in the clinical trial described herein.

As reported in this manuscript, the addition of CpG 10104 to the aluminum hydroxide adjuvant formulation of *Na*-GST-1 resulted in approximately a ten-fold significant increase in antigen-specific IgG antibodies as quantified by a qualified ELISA, compared to *Na*-GST-1/Alhydrogel without CpG 10104. This is in contrast to the use of a different TLR agonist, Glucopyranosyl Lipid A (GLA), which did not result in improved IgG levels to *Na*-GST-1 when added as an aqueous formulation to *Na*-GST-1/Alhydrogel [3]. GLA, a Th1-inducing synthetic TLR4-agonist adjuvant that is a non-toxic derivative of lipopolysaccharide, has led to improved immunogenicity of investigational vaccines for tuberculosis [15] and influenza [16]. This emphasizes that optimizing vaccine adjuvant formulations often requires empiric testing in humans.

The results of this Phase 1 trial raise several interesting possibilities that should be explored in future studies. Given the similar IgG responses seen with the CpG 10104 formulations of

the 30 µg and 100 µg doses of the *Na*-GST-1 antigen, both of which were significantly higher than the 100 µg dose of *Na*-GST-1/Alhydrogel without CpG 10104, the TLR9 immunostimulant may have maximized the humoral immune response to the vaccine. This suggests that CpG 10104 may allow for a lower dose of antigen in an eventual final vaccine product, resulting in a reduction in the cost of goods with respect to the recombinant protein component. Such an antigen-sparing effect by the use of a CpG ODN has been shown with other vaccines, including for an inactivated influenza vaccine for which 1/10th of a dose in combination with CpG 7909 showed superior immunogenicity to a full, unadjuvanted dose [17].

Not only did the addition of CpG 10104 result in similarly high IgG responses to *Na*-GST-1, but the level induced after the third dose was not much higher than after the second dose, although the lack of a statistically significant difference between these two time points may have been due to the small group sizes. Given this interesting finding, future studies should also explore whether the use of the CpG 10104 adjuvant might permit the use of a two-dose vaccination schedule instead of three doses, which would greatly improve uptake and lower costs of the vaccine. Such dose sparing with the addition of a CpG ODN has been demonstrated with the hepatitis B vaccine, where the seroprotection rate following two doses of Heplisav-B (recombinant hepatitis B surface antigen plus the CpG 1018 immunostimulatory sequence) was significantly higher than after three doses of Engerix-B (recombinant HBsAg formulated on aluminum hydroxide adjuvant) [18], leading to the licensure of this product as a two-dose regimen.

In contrast to overall IgG, IgG1, and IgG3 responses, anti-*Na*-GST-1 IgG4 levels were highest in the vaccine formulation that did not include CpG, indicating a potential shift towards $T_H1$ response by the addition of this TLR9 agonist. This shift may indeed be beneficial given that repeated vaccination may induce IgG4 antibodies that can interfere with the action of protective antibodies, as has been shown with vaccines being developed for HIV [19] and malaria [20]. Most recently, immune tolerance to the SARS-CoV-2 Spike protein was associated with IgG4 antibodies induced by repeated vaccination with an mRNA COVID-19 vaccine [21]. Since higher levels of IgG4 antibodies have been associated with increased COVID-19 mortality [22,23], advancing vaccine formulations such as the CpG 10104 formulation in the current study that do not result in induction of antigen-specific IgG4 may be preferred.

The improvement in the magnitude, and potentially the quality, of the humoral immune response to *Na*-GST-1 resulting from the addition of the CpG ODN, fortunately, did not come at the expense of safety. Concern had been raised early in the development of CpG ODN adjuvants that they might induce autoreactive B cells and thus increase the risk of autoimmune disease [24]. In particular, since CpG ODNs are derived from bacterial DNA, the risk of inducing autoantibodies to human DNA, and therefore the potential complication of systemic lupus, has been raised. Reassuringly, in the study reported herein, no recipients of the CpG 10104 formulation of the vaccine developed antibodies to dsDNA, ANA, or rheumatoid factor, and none reported any symptoms concerning for autoimmune disease. The mild, transient decreases in ANC seen in two recipients of the 100 µg dose of *Na*-GST-1/Alhydrogel plus CpG 10104 are similar to those reported in other studies that included a CpG ODN vaccine adjuvant [25]. No infections or other clinical evidence for immune suppression occurred in conjunction with these transient episodes of neutropenia. This effect is hypothesized to be due to the redistribution of neutrophils from the circulation to lymphoid tissues [26].

Although we have demonstrated the potent effect of the CpG 10104 adjuvant on the immunogenicity of *Na*-GST-1/Alhydrogel, one potential limitation to including it in an eventual licensed hookworm vaccine is the additional cost of goods for vaccine manufacture. However, this could be offset by the savings realized by the antigen and dose sparing that the use of the CpG ODN might produce as outlined above. Indeed, the recent example of the Corbevax

COVID-19 vaccine is encouraging, given that this vaccine manufactured by an Indian company was sold for less than $2 a dose.

An additional potential concern is that although anti-*Na*-GST-1 IgG antibody levels remained significantly higher than baseline at the end of the study, a steady decrease from peak levels was observed after the third vaccinations. Whether continued decay in IgG levels occurs, and if additional booster doses of the vaccine will be required to maintain sufficient humoral immune responses, will need to be explored in future, longer-term studies. However, it is possible that administration of the vaccine to individuals, including children, in hookworm-endemic areas may yield different immune response results, including the possibility that natural exposure to the parasite may boost anti-*Na*-GST-1 IgG responses, thus obviating the need for additional vaccine doses to maintain sufficient immunity.

The *Na*-GST-1/Alhydrogel vaccine targets only one of the two major human hookworm parasites. However, due to the high degree of sequence homology between *N. americanus* GST-1 and that of *Ancylostoma duodenale*, the other major human hookworm parasite, estimated to be from 55% to over 75% [27], we hypothesize that immunization with *Na*-GST-1 might not only protect humans against *N. americanus* infection, but possibly against other hookworm infections including *A. duodenale*. However, this will have to be formally tested in field efficacy or controlled infection clinical trials.

In conclusion, the Phase 1 trial reported herein confirms the hypothesis that the addition of the CpG 10104 TLR9 agonist to the *Na*-GST-1/Alhydrogel hookworm vaccine significantly increases the humoral immune response to the vaccine antigen without compromising safety or tolerability. Additional testing of this vaccine formulation has therefore been initiated in an endemic setting in participants who have been exposed to *N. americanus* (NCT03373214), as well as in a controlled human infection study (NCT03172975) to assess if the improved IgG response observed with the addition of CpG 10104 translates into a positive impact on vaccine efficacy against infection.

## Supporting information

**S1 Table. Number and percentage of participants experiencing unsolicited related adverse events by maximum severity and vaccine group.**
(DOCX)

**S2 Table. Clinical laboratory adverse events by vaccine group after any dose.**
(DOCX)

## Acknowledgments

We sincerely thank the study volunteers for their participation during the trial, as well as Judy Falloon, Shital Patel, Larissa May, and Matthew Laurens for their contributions as members of the study safety monitoring committee.

## Author contributions

**Conceptualization:** David J. Diemert, Doreen Campbell, Maria Elena Bottazzi, Peter J. Hotez, Jeffrey M. Bethony.

**Data curation:** David J. Diemert, Maria Zumer, Christina Gibbs-Tewary, Elissa M. Malkin, Doreen Campbell, Lara Hoeweler, Guangzhao Li, Jeffrey M. Bethony.

**Formal analysis:** David J. Diemert, Mark Bova, Elissa M. Malkin, Guangzhao Li, Jeffrey M. Bethony.

**Funding acquisition:** David J. Diemert, Maria Elena Bottazzi, Peter J. Hotez, Jeffrey M. Bethony.

**Investigation:** David J. Diemert, Maria Zumer, Lara Hoeweler, Guangzhao Li, Jeffrey M. Bethony.

**Methodology:** David J. Diemert, Jeffrey M. Bethony.

**Project administration:** David J. Diemert, Maria Zumer, Elissa M. Malkin, Doreen Campbell, Maria Elena Bottazzi, Peter J. Hotez, Jeffrey M. Bethony.

**Resources:** Maria Elena Bottazzi, Peter J. Hotez.

**Supervision:** David J. Diemert, Maria Elena Bottazzi, Peter J. Hotez, Jeffrey M. Bethony.

**Visualization:** David J. Diemert, Mark Bova, Christina Gibbs-Tewary, Lara Hoeweler, Guangzhao Li.

**Writing – original draft:** David J. Diemert, Mark Bova, Christina Gibbs-Tewary, Jeffrey M. Bethony.

**Writing – review & editing:** Maria Zumer, Elissa M. Malkin, Doreen Campbell, Guangzhao Li, Maria Elena Bottazzi, Peter J. Hotez.

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
