## [Decision Letter · Decision Letter 0]

11 Nov 2024

Response to Reviewers
Revised Manuscript with Track Changes
Manuscript

Shaden Kamhawi

co-Editor-in-Chief

Paul Brindley

co-Editor-in-Chief

**Journal Requirements:**
**Additional Editor Comments (if provided):**
**Reviewers' comments:**

**Key Review Criteria Required for Acceptance?**

**Methods**

-Are the objectives of the study clearly articulated with a clear testable hypothesis stated?

-Is the study design appropriate to address the stated objectives?

-Is the population clearly described and appropriate for the hypothesis being tested?

-Is the sample size sufficient to ensure adequate power to address the hypothesis being tested?

-Were correct statistical analysis used to support conclusions?

-Are there concerns about ethical or regulatory requirements being met?

Reviewer #1: see below

Reviewer #2: The study design is consistent with the objectives of the study.

**Results**

-Does the analysis presented match the analysis plan?

-Are the results clearly and completely presented?

-Are the figures (Tables, Images) of sufficient quality for clarity?

Reviewer #1: see below

Reviewer #2: All data are clearly presented.

**Conclusions**

-Are the conclusions supported by the data presented?

-Are the limitations of analysis clearly described?

-Do the authors discuss how these data can be helpful to advance our understanding of the topic under study?

-Is public health relevance addressed?

Reviewer #1: see below

Reviewer #2: The conclusions are supported by the data, and the discussion is quite complete.

**Editorial and Data Presentation Modifications?**

Reviewer #1: minor revision

Reviewer #2: (No Response)

**Summary and General Comments**

Reviewer #1: Nice study

Randomized observer blind Phase 1 analysis effectively evaluating impact of addition of a new CpG 10104 TLR9 adjuvant

Underlying vaccine Na-GST-1 with alum (anti hookworm candidate)

Hope was to increase Ab responses (IgG)

Various hookworm vaccines based on Na-GST-1 and or Na-APR-1 have been in various human studies, including with a TLR4 adjuvant GLA

No knowledge of what IgG level might be protective, if at all

Proposed mechanism is that anti-hookworm antibodies might impact feeding, toxic heme handling, and thus perhaps attachment/fecundity/duration/feeding etc, resulting in less disease ant individual and or community level

The weakness of not knowing what if any level of antibody might be needed is an inherent weakness, but can be a addressed if/when the vaccines advance

The assumption that more is better is reasonable for now

Unfortunately, it will be hard even with a great vaccine to maintain durable high antbody levels, and since the antigens are in the worms intestine, infection wouldn’t be expected to induce a booster or recall response

These are challenging issues

It is in this context that the current study is performed

The vaccine itself does not appear very immunogenic (it is true values increased and responder frequencies increased; but there was only a single log10 increase in serum levels after 3 IM inoculations (spaced for maximal effect every 2 months) of 100 mcg (big dose) of the vaccine with alum (already one adjuvant that assist Th2 Ab responses).

The CpG 10104 impact improves this and thus looks very nice, and that is the main finding

The data that 2 doeses with CpG appears to plateau is promising, as is a comparable response at 30 and 100 mcg vaccine, as long as CpG 10104 included

I would ask the authors to give more background on CpG 10104 (this is effectively a 10104 study)

Is this first in human?

If so, why no adjuvant alone group?

If not, please describe and reference prevous work.

What animal studies have been done with 10104

How about toxicology?

Although looks safe, it is of note that with only 16 people getting 10104, one had tenderness at injection site that lasted for a month, and another person was classified as a “severe” local reaction

Would suggest stating that in discussion and say this will need to be followed in future studies

The authors give references for other CpGs, but the pertinent issue here is 10104, so please give more details.

Also

Please discuss what % Hw Na vs Ad

Is GST-1 Conserved?

Would an anti-GST protect against both, or are we only going after 1 of the 2 main global causes even if all works well

Please help reader with anti-GST background: has animal variants shown protection in their appropriate animal models (example dog or cat or other); if that work not done, would state.

Although CpG helps, as expected there is a decline in IgG that would kinetically map (even after after 3 doses with two adjuvants and high dose vaccine) to being back to baseline in 18-24 months; would at least point that out in discussion since even if can get by with 2 initial doses, this strategy may be looking at yearly or q24 month boosting to maintain meaningful Ab levels; this would be difficult with kids in LMIC setting since not EPI, although could be combined with annual MDAs; would discuss

Please give CTN for the endemic zone study that the paper references twice and says has been initiated

Can the authors more clearly state how many Hw vaccines are in development and their stage? Formulations and different teams are fluid, and would help reader.

Overall, important work that is encouraging that 10104 assists.

Reviewer #2: The study by Diemert et al, present safety results from a vaccination with GST-1 as an anti-hookworm vaccine. The novelty reside in the investigation of CpG as a novel adjuvant to boost antibody production. Interestingly, the authors show that Alum +CpG do boost antibody production and is well tolerated. A question of interest to answer in the future would be whether this increase in antibody is also reflected in an increase of neutralizing antibody. The authors could consider an in vitro assays with NEcator americanus or Nippostrongylus brasiliensis as a surrogate and study the parasite viability in presence of blood and increasing concentration of serum isolated from vaccinated patients.

PLOS authors have the option to publish the peer review history of their article (what does this mean? ). If published, this will include your full peer review and any attached files.

**Do you want your identity to be public for this peer review?** For information about this choice, including consent withdrawal, please see our Privacy Policy .

Reviewer #1: No

Reviewer #2: No

**Figure resubmission:****Reproducibility:** To enhance the reproducibility of your results, we recommend that authors of applicable studies deposit laboratory protocols in protocols.io, where a protocol can be assigned its own identifier (DOI) such that it can be cited independently in the future. Additionally, PLOS ONE offers an option to publish peer-reviewed clinical study protocols. Read more information on sharing protocols at https://plos.org/protocols?utm_medium=editorial-email&utm_source=authorletters&utm_campaign=protocols

---

## [Editor Report · Decision Letter 1]

16 Dec 2024

Dear Dr. Diemert,

We are pleased to inform you that your manuscript 'Randomized, observer-blind, controlled Phase 1 study of the safety and immunogenicity of the Na-GST-1/Alhydrogel hookworm Vaccine with or without a CpG ODN adjuvant in hookworm-naïve adults' has been provisionally accepted for publication in PLOS Neglected Tropical Diseases.

Best regards,

Michael Cappello

Academic Editor

Francesca Tamarozzi

Section Editor

Shaden Kamhawi

co-Editor-in-Chief

Paul Brindley

co-Editor-in-Chief

---

## [Editor Report · Acceptance letter]

Dear Dr. Diemert,

We are delighted to inform you that your manuscript, "Randomized, observer-blind, controlled Phase 1 study of the safety and immunogenicity of the Na-GST-1/Alhydrogel hookworm Vaccine with or without a CpG ODN adjuvant in hookworm-naïve adults," has been formally accepted for publication in PLOS Neglected Tropical Diseases.

Best regards,

Shaden Kamhawi

co-Editor-in-Chief

Paul Brindley

co-Editor-in-Chief
